# Neuronal cell fate diversification controlled by sub-temporal action of *Kruppel*

**Johannes Stratmann[1†], Hugo Gabilondo[1,2†], Jonathan Benito-Sipos[2], Stefan Thor[1]***

[1]Department of Clinical and Experimental Medicine, Linköping University, Linköping, Sweden; [2]Departamento de Biología, Universidad Autónoma de Madrid, Madrid, Spain

**Abstract** During *Drosophila* embryonic nervous system development, neuroblasts express a programmed cascade of five temporal transcription factors that govern the identity of cells generated at different time-points. However, these five temporal genes fall short of accounting for the many distinct cell types generated in large lineages. Here, we find that the late temporal gene *castor* sub-divides its large window in neuroblast 5–6 by simultaneously activating two cell fate determination cascades and a sub-temporal regulatory program. The sub-temporal program acts both upon itself and upon the determination cascades to diversify the *castor* window. Surprisingly, the early temporal gene *Kruppel* acts as one of the sub-temporal genes within the late *castor* window. Intriguingly, while the temporal gene *castor* activates the two determination cascades and the sub-temporal program, spatial cues controlling cell fate in the latter part of the 5–6 lineage exclusively act upon the determination cascades.

***For correspondence:** stefan. thor@liu.se

[†]These authors contributed equally to this work

**Competing interests:** The authors declare that no competing interests exist.

## Introduction

During nervous system development, neural progenitor cells often undergo stereotyped changes in their competence, evident by the sequential and programmed generation of distinct neuronal and glial sub-types (*Okano and Temple, 2009*; *Pearson and Doe, 2004*). In the *Drosophila* embryonic central nervous system (CNS), neuroblasts (NBs) sequentially expresses the transcription factors, Hunchback (Hb) > Kruppel (Kr) > POU-homeodomain factors Nubbin and Pdm2 (Pdm) > Castor (Cas) > Grainy head (Grh) (*Baumgardt et al., 2009*; *Brody and Odenwald, 2000*; *Isshiki et al., 2001*; *Novotny et al., 2002*). These factors temporally alter NB competence to determine the types of neurons and glia born at each step of lineage progression (*Kohwi and Doe, 2013*; *Li et al., 2013*). However, because *Drosophila* NB lineages can generate an array of different cell types, the instructive capacity of five temporal genes falls short of explaining the diversity observed (*Baumgardt et al., 2009*; *Tsuji et al., 2008*). Studies suggest that this regulatory challenge is solved by the activity of the so-called sub-temporal genes, which act in cascades downstream of the temporal genes, do not feedback on the temporal genes, and play a role in sub-dividing larger temporal competence windows (*Baumgardt et al., 2009*; *Benito-Sipos et al., 2011*). Downstream of temporal cues, the specification of cell fate is subsequently controlled by determination genes, referred to as terminal selector genes, that activate repertoire(s) of terminal cell fate genes e.g., neurotransmitters and ion channels (*Hobert, 2008*; *Wenick and Hobert, 2004*). The terminal selectors have been found to often act in combinatorial codes to dictate final and unique cell fate (*Allan and Thor, 2015*; *Baumgardt et al., 2007*; *Enriquez et al., 2015*; *Sharma et al., 1998*; *Thor et al., 1999*). In addition, terminal selectors may act in cascades denoted coherent feedforward loops (FFLs) (*Mangan and Alon, 2003*; *Mangan et al., 2003*). FFLs are common in *E.coli* and yeast gene regulatory networks

**eLife digest** As a nervous system develops, stem cells generate different types of nerve cells at different times. This series of events follows a fixed schedule in developing embryos, and even a single stem cell that is removed and then grown outside the body will follow the same schedule. This strongly suggests that stem cells have a built-in clock that controls their development.

Studies of the developing nervous system of fruit flies reveal that this clock works by switching genes on in specific sequences, which defines which nerve cells are produced at different stages of development. However, a clock built from the genes that are currently known to be involved in the process is simply not fine-tuned enough to explain how so many different types of nerve cell develop at such precise times. This implies that scientists do not yet know all of the genes that are involved.

Using genetic experiments in stem cells from fruit flies, Stratmann, Gabilondo et al. now identify additional clock genes that act to divide broad windows of time during development into smaller, more precise ones. Genes that define broad windows of time switch on the "small window" genes at specific times – a bit like large cogs turning small cogs in a clock. One small window gene, called *Kruppel*, works at different stages of development and it is possible that other small window genes multi-task in similar ways in other developmental clocks, such as those found in more complex organisms like humans.

It is clear that many genes work in sequence in the developing nervous system to ensure that developmental stages happen at precise times. Stratmann, Gabilondo et al. will next investigate the molecular details of this timing, specifically how genes in sequential time windows connect together like cogs in the developmental clock.

(*Alon, 2007*), but have also been identified in animals, including in both *Drosophila* and *C.elegans* (*Baumgardt et al., 2009*; *Baumgardt et al., 2007*; *Etchberger et al., 2009*; *Johnston et al., 2006*). However, how temporal and sub-temporal genes intersect with terminal selector FFLs to dictate cell fate is poorly understood.

The Apterous (Ap) neurons of the *Drosophila* ventral nerve cord (VNC) constitute a group of inter-neurons expressing the LIM-HD factor Apterous (Ap) (*Lundgren et al., 1995*). Because of a multi-tude of antibody markers and genetic tools available for Ap neurons, these cells have been subject to a number of studies of cell fate specification. Ap neurons can be subdivided into; (1) dorsal Ap neurons (dAp) that are a dorsal bi-lateral row of Ap neurons generated in abdominal and thoracic segments by NB4-3, and (2) the Ap cluster that are a bi-lateral group of four Ap neurons, denoted Tv1-Tv4, that are generated consecutively by NB5-6T in thoracic segments (*Figure 1*) (*Baumgardt et al., 2007*; *Gabilondo et al., 2016*; *Park et al., 2004*). Two out of four Ap cluster cells have a neuropeptidergic cell fate; the Tv1/Nplp1 and Tv4/FMRFa cells (*Baumgardt et al., 2007*; *Benveniste et al., 1998*; *Park et al., 2004*), while Tv2 and Tv3 are Ap interneurons. All four cells express Ap and the transcriptional co-factor Eyes absent (Eya) (*Miguel-Aliaga et al., 2004*). Two related terminal selector FFLs operate in Ap cluster cells to dictate Nplp1 or FMRFa cell fate, *col>ap/eya>dimm>Nplp1* and *ap/eya/dac>dimm/BMP>FMRFa* (*Allan et al., 2005, 2003*; *Baumgardt et al., 2007*; *Miguel-Aliaga et al., 2004*). Each cell type-specific FFL cascade is triggered by specific temporal and spatial inputs established during lineage progression. The spatial input, conferred by body position, consists of the combinatorial action of the Hox homeotic gene *Antennapedia* (*Antp*), the Hox co-factors *extradenticle* (*exd*) and *homothorax* (*hth*), as well as the homeobox gene *ladybird early* (*lbe*) (*Gabilondo et al., 2016*; *Karlsson et al., 2010*). The temporal input is mediated by Cas and Grh; the last two factors in the Hb>Kr>Pdm>Cas>Grh temporal cas-cade (*Baumgardt et al., 2009*). Initially, these spatial and temporal inputs activate the two terminal FFL cascades in all four Ap cluster neurons. However, Cas triggers several additional genes in the neuroblast NB5-6T that act to sub-divide the broader Cas window. These genes were denoted sub-temporal genes, and consist of *squeeze* (*sqz*), *nab* and *seven up* (*svp*), which act to repress *collier* (*col*; Flybase *knot*) in the generic Tv2/3 neurons and the Tv4 cell (*Baumgardt et al., 2009*; *Benito-Sipos et al., 2011*). The repression of *col* in the Tv2/3 and Tv4 neurons prevents those cells from

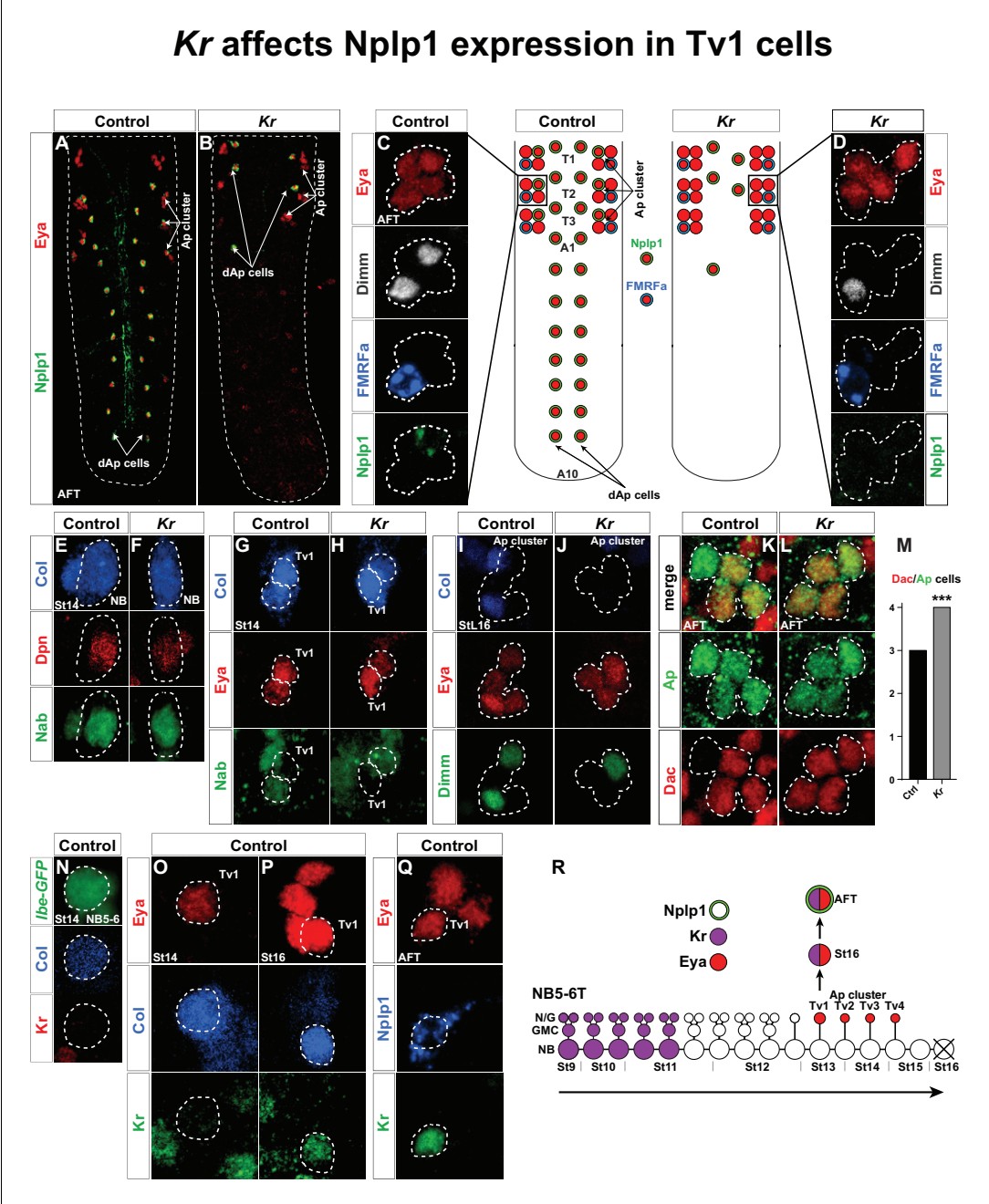

**Figure 1.** *Kr* affects Nplp1 expression in Tv1 cells. (**A–B**) Whole VNCs of control and *Kr* mutants, at AFT, reveal loss of Nplp1 expression in the dAp cells, but also in the Tv1 cells. (**C–D**) Ap cell clusters at AFT, showing an expression of Eya, Dimm, FMRFa and Nplp1, in control (**C**) and *Kr* mutants (**D**). In mutants, while Eya is normally expressed in four cells, Dimm and Nplp1 expression is lost in the Tv1 cell. Dimm and FMRFa expression in the Tv4 cell is not affected in *Kr* mutants compared to control. (**E–F**) Nab and Col expression, in the NB5-6T at St14, is similar in control and *Kr* mutants. (**G–H**) In Tv1 cells (dashed circles) at St14 (distinguishable from Tv2 cells by positive Eya but negative Nab expression) Col expression is similar in control and *Kr* mutants. (**I–J**) At late St16, while Eya is un-affected, Col and Dimm expression is lost in the Tv1 cell in *Kr* mutants. (**K–L**) While Dac is usually expressed in three out of four cells in the Ap clusters at AFT, we find that in *Kr* mutants Dac expression is observed in all four Ap cluster cells. (**M**) Dac is significantly upregulated in *Kr* mutants when compared to control (***$p \leq 0.001$, $n_{ctrl}$ = 7 clusters, $n_{Kr}$ = 10 clusters, Chi-square test). (**N–Q**) Timeline of Kr expression in control shows late onset of Kr expression specifically in the Tv1 cell. (**N**) Kr is not expressed in the NB5-6T or (**O**) the Tv1 cell at St14. (**P**) At St16, Kr expression commences in Tv1 (dashed circle), which is characterized by maintenance of strong Col expression. (**Q**) Variably, Kr expression in Tv1 is maintained into AFT, identifiable by co-staining for Eya and Nplp1. (**R**) Model of NB5-6T, showing the early Kr expression commencing at St9 and persisting until St11, together with the Eya positive postmitotic Ap cluster cells, out of which Tv1 starts to express Kr at St16, to ultimately specify into the Nplp1 positive Tv1 cells. Genotypes: (**A, C, E, H, J, L, O, P, Q**) *OregonR*. (**B, D, F, I, K, M**) *Kr¹, KrᶜᴰD*. (**N**) *lbe(K)-EGFP*.

being specified into Tv1/Nplp1 neurons. However, in spite of the identification of the three sub-temporal genes *sqz*, *nab* and *svp*, the precision of Ap cluster cell specification clearly indicated the existence of additional players.

In a recent study, we identified *Kr* as being important to trigger a final cell fate specification cascade in the neuropeptidergic dAp neurons. We found that *Kr* acts in a predicted and typical early temporal role in the NB4-3 lineage, to initiate the final dAp specification (*Gabilondo et al., 2016*). However, upon further analysis of *Kr* mutants we found that the neuropeptidergic cell fate of the Nplp1 cell in the Ap cluster generated by NB5-6T is also lost. Here, we find that *Kr* mutants show a specific loss of Col, Dimm and Nplp1 expression in the Tv1 cell, while specification of the other three Ap cluster neurons is unaffected. We find that Kr is expressed in a second phase in the NB5-6T lineage; postmitotically in the Tv1/Nplp1 cell. The late temporal gene *cas* activates *Kr* in Tv1, while the sub-temporal genes *sqz* and *nab*, which are also activated by *cas*, repress *Kr*. In contrast, *Kr* does not repress *sqz* or *nab*, but rather acts to suppress *svp* in the Tv1 cell, thereby preventing *svp* from repressing Col and Dimm. Hence, an initial generic Ap cluster cell fate, triggered by *cas* and spatial cues, is sub-divided by four sub-temporal genes, also activated by *cas*, that act on each other and on the two terminal selector FFLs.

This study reveals that the sub-division of a broader temporal competence window is controlled by an elaborate sub-temporal regulatory program triggered by one and the same temporal factor, which also triggers the terminal selector FFLs. It furthermore demonstrates that a member of the canonical *Drosophila* temporal cascade can act at two stages of a lineage. Finally, it shows that sub-temporal genes can act both on the terminal selector FFLs and on each other.

## Results

### *Kruppel* is expressed postmitotically in Tv1 neurons and controls their terminal specification

We recently identified *Kruppel (Kr)* as being important for the development of the NB4-3 lineage, and for specification of the dAp/Nplp1 neurons generated in this lineage (*Gabilondo et al., 2016*). However, upon analyzing *Kr* mutants we noted that *Kr* not only affects Nplp1 expression in dAp cells, but also in the Tv1 cells of the Ap cluster (*Figure 1A–D*). To begin addressing the role of *Kr* in Tv1 specification, we analyzed the expression of several Tv1 specification factors: Collier (Col; an EBF/COE factor), Ap (LIM-HD), Eyes absent (Eya; nuclear phosphatase) and Dimmed (Dimm; bHLH) (*Allan et al., 2005*; *Baumgardt et al., 2007*; *Hewes et al., 2003*; *Miguel-Aliaga et al., 2004*; *Park et al., 2004*). While Ap and Eya were unaffected, Dimm expression was lost from Tv1 (*Figure 1C–D*). In contrast, Dimm and FMRFa expression in the Ap4 cell was unaffected (*Figure 1C–D*). With regards to Col, expression was unaffected at the onset of expression in the NB and in Tv1/Ap1 (*Figure 1E–H*). However, by stage 16 (St16) we noted that Col was lost from the Tv1 cell (*Figure 1I–J*). We also analyzed expression of the transcriptional co-factor Dachshund (Dac), important postmitotically for Tv4/FMRFa fate (*Miguel-Aliaga et al., 2004*). Dac is normally expressed in Tv2, Tv3 and Tv4, but in *Kr* mutants Dac expression expands into the Tv1 cell (*Figure 1K–L*). The sub-temporal factor Nab (*Baumgardt et al., 2009*) was not affected in Kr mutants (*Figure 1E–H*).

*Kr* acts at an early stage in the canonical temporal gene cascade (*Isshiki et al., 2001*; *Kohwi and Doe, 2013*), and was found to be expressed during St9-St11 in NB5-6T (*Baumgardt et al., 2009*) (*Figure 1R*). We were therefore intrigued by the late and restricted specification phenotype in the Tv1 cells observed in *Kr* mutants. These effects prompted us to analyze the expression of Kr in the NB5-6T neuroblast at St14 and in the Ap cluster at St14 onward to late embryonic stage (air-filled trachea stage; AFT). In agreement with previous Kr expression analysis (*Baumgardt et al., 2009*), we did not detect Kr expression in the NB5-6T or in the Tv1 cell at St14 (*Figure 1N–O*). However, by St16 *Kr* expression was evident in the Tv1 cell (*Figure 1P*). With some variability, *Kr* expression is maintained into stage AFT, and overlaps with the onset of Nplp1 expression (*Figure 1Q*).

In summary, Kr shows two phases of expression in the NB5-6T lineage; an early phase, in line with its known role as an early temporal factor, and a postmitotic phase in the Tv1 neuron (*Figure 1R*). We observe highly selective effects on the Ap cluster specification in *Kr* mutants, specifically affecting the Tv1 neuron; normal Nab and Col expression in the NB, normal Eya and Ap expression in the Ap cluster cells, ectopic Dac expression in Tv1, subsequent loss of Col expression in Tv1, and failure

to turn on Dimm and Nplp1 expression in Tv1, while Dimm and FMRFa expression in Tv4 is unaffected.

## *Kr* is sufficient to govern Tv1/Nplp1 cell fate

The selective expression of Kr in the postmitotic Tv1 neuron, and the importance of Kr for Tv1 specification, prompted us to ectopically express *Kr*, by crossing *UAS-Kr* to the late Ap neuron specific driver *ap-Gal4* (*Allan et al., 2005*). Misexpression of *Kr* in all four postmitotic Ap neurons resulted in the ectopic expression of Col, Dimm and Nplp1 in other Tv neurons (*Figure 2A–D*). This is in line with previous findings (*Baumgardt et al., 2007*), showing that maintained expression of Col in all four Ap cluster neurons results in ectopic Dimm and Nplp1 expression. The conversion of Ap cluster

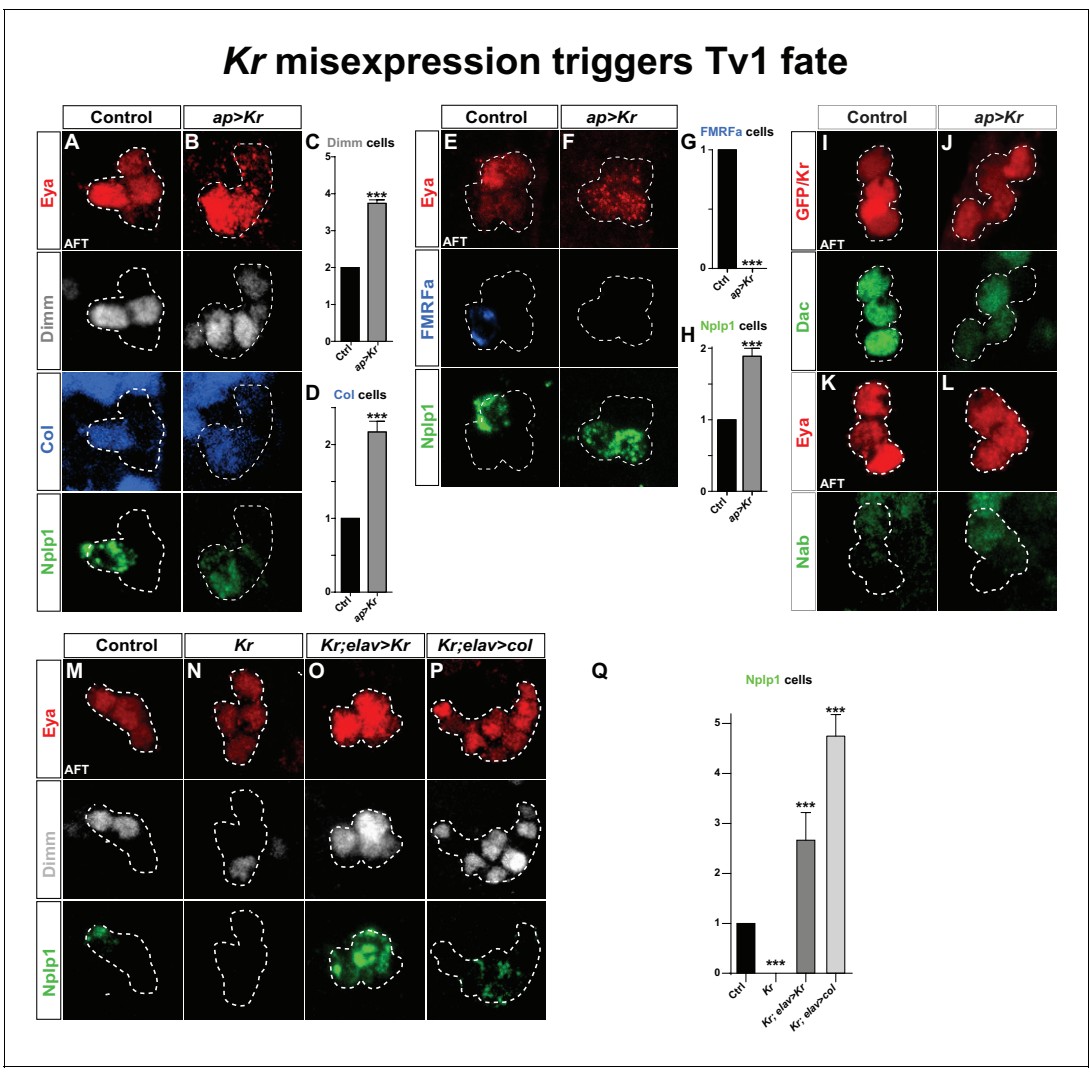

**Figure 2.** *Kr* misexpression triggers Tv1 fate. (A–D) Postmitotic misexpression of *UAS-Kr* driven by *ap-Gal4* results in ectopic expression of Dimm, Col and Nplp1 in the Ap cluster cells. (C–D) Quantification of Dimm and Col cells in the Ap cluster shows a significant increase when *Kr* is misexpressed (***p≤0.001; n ≥ 12 clusters; Chi-square test +/- SEM). (E–H) Kr misexpression triggers ectopic expression of Nplp1 in the Ap cluster, but loss of FMRFa expression. (G–H) Quantification of FMRFa and Nplp1 expressing cells in the Ap cluster (***p≤0.001; n ≥ 9 clusters; Chi-square test +/- SEM). (I–L) Misexpression of *Kr*, does not affect Dac or Nab expression. (M–O) *Kr* rescue by *UAS-Kr* driven from *elav-Gal4* restores Dimm and Nplp1 specifically to Tv1. (P) Cross-rescue of *Kr* with *UAS-col* from *elav-Gal4* rescues *Kr* mutants, evident by Dimm and Nplp1 expression in the Ap clusters. In both rescue and cross-rescue, driving each *UAS* transgene from *elav-Gal4* triggers supernumerary Tv1 cells. (Q) Quantification of the number of cells expressing Nplp1 in the Ap cluster (***p≤0.001; n ≥ 9 clusters; Chi-square test +/-SEM). Genotypes: (A, E, K, M) *OregonR*. (B, F, J, L) *ap-Gal4/UAS-Kr; +/UAS-Kr.* (I) *UAS-GFP/ap-Gal4.* (N) *Kr[1], Kr[CD].* (O) *elav-Gal4/+; Kr[1], Kr[CD]; UAS-Kr/+.* (P) *Kr[1], Kr[CD]; UAS-col/elav-Gal4.*

neurons to a Tv1 cell fate was accompanied by the loss of FMRFa expression (*Figure 2E–H*), but not by the loss of Dac or Nab expression (*Figure 2I–L*).

Because Col is essential for activating Dimm and Nplp1, the loss of their expression in *Kr* mutants may be a direct consequence of the loss of Col expression. To test this idea, we attempted to 'cross-rescue' *Kr* with *UAS-col*. First, as a control experiment, we attempted to rescue *Kr* with *UAS-Kr*, using the *elav-Gal4* driver line, which is expressed in the NB5-6T at St12 and onward (*Karlsson et al., 2010*). Indeed, we noted robust rescue of Dimm and Nplp1 in the Ap clusters, and as anticipated from the Kr misexpression experiments (above) there were supernumerary Tv1 neurons (*Figure 2M–O, 2Q*). Next, we cross-rescued *Kr* with *UAS-col*, and observed rescue of Tv1 neurons, evident by the expression of Dimm and Nplp1 (*Figure 2P,Q*). As previously, described (*Baumgardt et al., 2007*), ectopic expression of *col* from *elav-Gal4* resulted in supernumerary Eya, Dimm and Nplp1 cells (*Figure 2P,Q*).

These results suggest that the role of *Kr* in the Tv1 neuron is not to act directly upon *Nplp1*, but rather to ensure the maintenance of Col, and thereby ensure the propagation of the *col>ap/eya>-dimm>Nplp1* feedforward cascade leading to Tv1/Nplp1 terminal differentiation.

## Postmitotic Kr expression is activated by the late temporal gene *castor*

The selective expression of Kr in the Tv1 cell is key for the discriminated specification of the Tv1 cell fate from that of the other Ap cluster cells. But what activates Kr exclusively in the Tv1 neuron? To address this, we first analyzed Kr expression in *ap, eya* and *col* mutants, critical Tv1 cell fate determinants (*Baumgardt et al., 2007*), but did not observe any change in Kr expression in these mutant backgrounds (*Figure 3—figure supplement 1*).

Next, we analyzed *ladybird early (lbe)* and *Antp* mutants, both of which are critical for NB5-6 identity (*lbe*) and for Ap cluster development (both) (*Gabilondo et al., 2016*; *Karlsson et al., 2010*). Since *Antp* and *lbe* show a loss of the Ap cluster markers Eya, Ap, Col, Dimm and Nplp1, we identified the NB5-6T lineage based on the specific expression of reporter genes under the control of an enhancer fragment from the *ladybird early* gene (*lbe(K)*) (*Figure 3—figure supplement 1*) (*Baumgardt et al., 2007*; *De Graeve et al., 2004*). In order to accurately monitor the Ap cluster cells in control, *Antp* and *lbe* mutant backgrounds, we used the non-affected Ap cluster markers Nab and Cas to thereby identify the Tv1 cells (*Figure 3—figure supplement 1*). At St16, when Kr is robustly expressed in Tv1, Nab is expressed in three out of four Ap cells and overlaps with Cas expression in Tv2/3 (*Baumgardt et al., 2009*). The Tv1 cell resides in immediate proximity to the two Cas/Nab double positive Ap cluster cells, and is identifiable by Kr and *lbe-GFP* expression, combined with the lack of Nab and Cas. Analysis of Kr expression in the Tv1 cell in *Antp* and *lbe* mutants, using this marker combination, revealed that Kr is not affected in either mutant (*Figure 3—figure supplement 1*).

Next, we addressed Kr expression in mutants for the *castor (cas)* temporal gene. *cas* plays a key role during NB5-6T lineage development, and regulates *col*, and thereby a number of Tv1 determinants, including Ap, Dimm and Nplp1 (*Baumgardt et al., 2009*). However, in *cas* mutants Eya is still expressed in the Ap cluster cells (*Baumgardt et al., 2009*). This allowed us to assess Kr expression in the Ap cluster cells within the NB5-6T lineage. We found that *cas* mutants often displayed loss of Kr in the Tv1 cells (*Figure 3A–C*). This was accompanied by a reduction in the number of Kr-expressing cells in the entire NB5-6T lineage (*Figure 3D*). The reduction of Kr expression in the entire NB5-6T lineage in *cas* mutants was mirrored by extensive overlap between Cas and Kr in the NB5-6T lineage in the wild type background (*Figure 3F*). Moreover, scanning the entire thoracic VNC we observed Cas/Kr overlap in a number of other cells (*Figure 3E*). Analyzing *cas* mutants in the entire thoracic segments, we observed striking reduction in Kr expression throughout these segments (*Figure 3G–H*).

Finally, we analyzed Kr expression in mutants for the last gene in the temporal gene cascade; *grainy head (grh)*, which is known to be expressed during the latter part of NB5-6T lineage development, and important for Tv4/FMRFa cell fate (*Baumgardt et al., 2009*). However, we did not observe any effect on Kr expression in *grh* mutants (*Figure 3—figure supplement 1*).

We conclude that Kr is activated in Tv1 by *cas*, and strikingly, that this regulatory connection extends into many late postmitotic cells in the VNC. In contrast, neither *Antp* nor *lbe* regulate Kr in the Tv1 neuron.

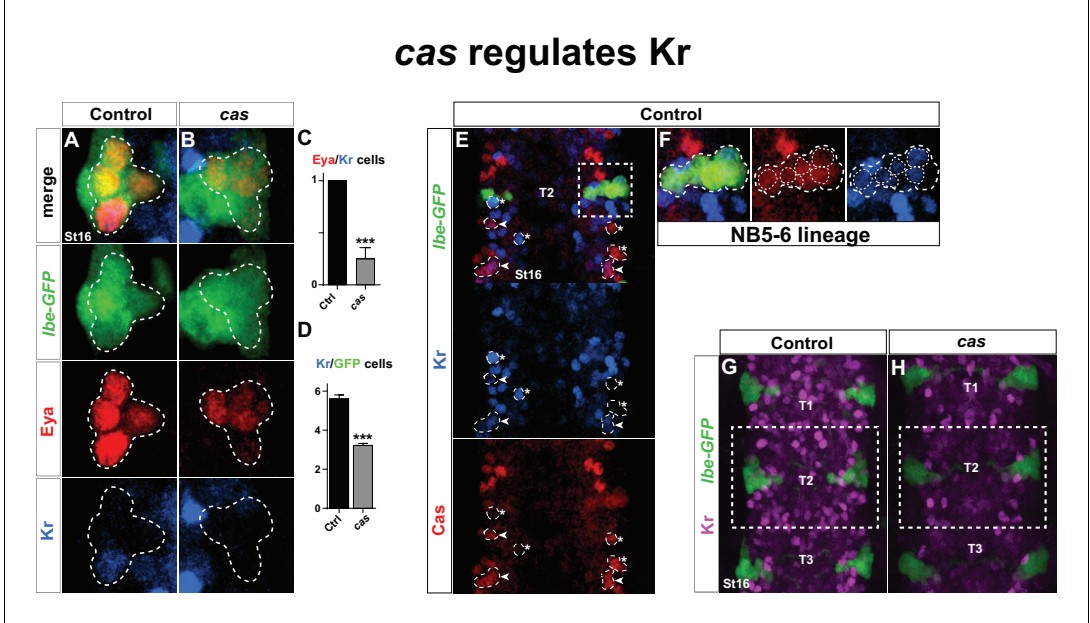

**Figure 3.** *castor* regulates Kr. (**A–B**) *Kr* expression in the Ap cluster cells is frequently lost in *cas* mutants at St16. Ap cluster cells were identified using Eya as a marker for the Ap cluster and *lbe(K)-eGFP* as a marker for the NB5-6T lineage. (**C–D**) Quantification of Eya/Kr cells and Kr/GFP cells in the Ap cluster and NB5-6T lineage, respectively (***p≤0.001, n ≥ 18 clusters, Chi-square test ± SEM, for (**C**), and Student's *t*-test ± SEM, for (**D**)). (**E–F**) Kr expression overlaps with Cas in the NB5-6T (**F**), as well as in many cells in the entire thorax (arrowhead marks overlapping cells, asterisk marks non-overlapping cells). (**G–H**) Kr expression is globally reduced in the thoracic region T1-T3 in *cas* mutants. Genotypes: (**A, K**) *lbe(K)-EGFP*. (**B, L**) *lbe(K)-EGFP /+; cas^{Δ3}/cas^{Δ1}*.

The following figure supplement is available for figure 3:

**Figure supplement 1.** Ap cluster cell fate determinants do not regulate Kr in Tv1.

## Kr expression is suppressed by the sub-temporal genes *squeeze* and *nab*

We find that Kr is specifically expressed in the Tv1 cell, and that Kr is necessary and sufficient, within the Ap cluster to specify Tv1 cell fate. Kr acts by ensuring maintenance of Col expression, thus allowing for the *col>ap/eya>dimm* feedforward cascade to progress and finally activate Nplp1 expression. We find that *cas* activates Kr in the Tv1 cell. However, Cas expression spans the latter half of the NB5-6T lineage, covering all four Ap neurons (*Baumgardt et al., 2009*), and hence does not readily explain Kr expression specifically in Tv1. Previous studies revealed that down-regulation of Col in Tv2, Tv3 and Tv4 is under control of a *cas>sqz>nab* FFL cascade, where down-regulation of Col is controlled by *sqz/nab* (*Baumgardt et al., 2009*). Because of the timed delay in the *cas>sqz>-nab* loop after an initial postmitotic phase (*Baumgardt et al., 2009*), this ensures that Col expression is only maintained in Tv1, and down-regulated in the other three Ap neurons. How does Kr relate to the *cas>sqz>nab* feedforward loop?

As described above, Nab is not affected in *Kr* mutants or by *Kr* misexpression (*Figures 1E–H*, *2K–L*). Thus, the *cas>sqz>nab* feedforward loop is not affected by *Kr*. In contrast, in both *sqz* and *nab* mutants, we observed one extra Kr/Dimm expressing cell in the Ap cluster (*Figure 4A–E*). While *sqz* is expressed by all four Ap cluster neurons, the timing-delay in the *cas>sqz>nab* loop results in Nab expression only in three latter born cells; Tv2, Tv3 and Tv4. Misexpression of *nab* suppresses the Col maintenance and results in loss of Dimm and Nplp1 from the Tv1 neuron (*Baumgardt et al., 2009*). In line with these findings, we addressed the effects of misexpressing *nab* in the Tv1 neuron, from *elav-Gal4*. We observed that *Kr* expression was absent from the Ap cluster in *nab* misexpression (*Figure 4K–M*).

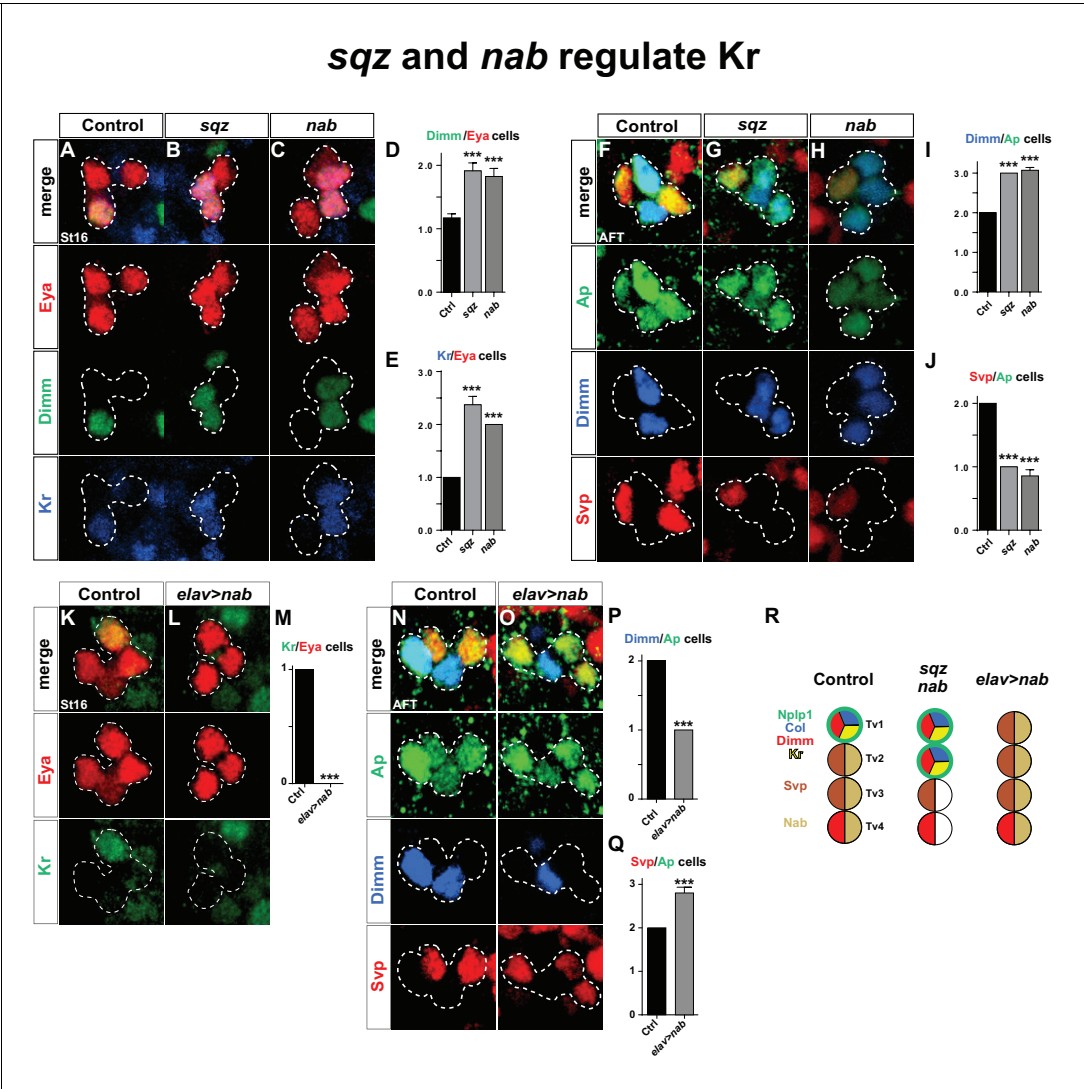

**Figure 4.** *squeeze* and *nab* regulate Kr. (**A–C**) At St16, both *sqz* and *nab* mutants show additional Dimm and Kr expressing cells in the Ap cluster. (**D–E**) Quantification of Dimm and Kr expressing Ap cluster cells in *sqz* and *nab* mutants (∗∗∗p≤0.001; n ≥ 17 clusters; Student's *t*-test +/-SEM for (**D**) and between Ctrl and *sqz* in (**E**); Chi-square test for Ctrl and *nab* in (**E**)). (**E–H**) Similar to St16, at AFT, both *sqz* and *nab* mutants show an increase in the numbers of Dimm expressing Ap cluster cells. (**I–J**) Quantification of Dimm/Ap and Svp/Ap cells (∗∗∗p≤0.001; n ≥ 14 clusters; Chi-square test; +/-SEM). (**K–L**) Misexpression of *nab* from *elav-Gal4* results in a significant decrease in Kr expressing Ap cluster cells, at St16. (**M**) Quantification of Kr/Eya Ap cluster cells (∗∗∗p≤0.001; n ≥ 16 clusters; Chi-square test). (**N–O**) *nab* misexpression from *elav-Gal4* at AFT shows a reduction of Dimm expressing and an increase in Svp expressing Ap cluster cells. (**P–Q**) Quantification of Dimm/Ap and Svp/Ap cluster cells (∗∗∗p≤0.001; n ≥ 10 clusters; Chi-square test; ± SEM). (**R**) Cartoon showing the Ap cluster in control, *sqz* and *nab* mutants, as well as *nab* misexpression, at AFT. Control Ap clusters show stereotyped expression of Nplp1, Col and Kr in the Tv1 cell. Dimm is present in both neuropetidergic Tv1/Nplp1 and Tv4/FMRFa cells. While Tv2-Tv4 cells all express *nab*, *svp* expression is restricted to Tv2 and Tv3. In *nab* and *sqz* mutants one of the 'generic' Tv2/Tv3 Ap neurons is converted into a Tv1/Nplp1 cells, by expression of *Kr*, which in turn suppresses *svp* expression, which allows for *col* and *dimm* expression. The misexpression of *nab* shows the reciprocal phenotype of the mutants, with one extra Svp expressing cell, loss of Kr, Col, Dimm and Nplp1. Genotypes: (**A, F, K, O**) *OregonR*. (**B, G**) *sqz^{ie}/sqz^{ie}*. (**C, H**) *nab^{R52}/nab^{R52}*. (**L, P**) *UAS-nab/+; elav-Gal4/+.*

We conclude that the sub-temporal genes *sqz* and *nab* act to restrict the expression of Kr to the Tv1 neuron, while, *Kr* does not reciprocally regulate the *cas>sqz>nab* feedforward loop.

## The main role of *Kr* is to suppress the sub-temporal gene *svp*

*Kr* does not repress the *cas>sqz>nab* feedforward loop, playing out in Tv2, Tv3 and Tv4. In addition, *Kr* can be cross-rescued by *col*, and hence is not critical for the *col>ap/eya>dimm>Nplp1*

feedforward cascade. What, then, is the mechanism by which *Kr* ensures maintenance of Col in Tv1 cells?

Previous studies found that *svp* mutants displayed ectopic Col, Dimm and Nplp1 expression in the Ap cluster, and that postmitotic misexpression of *svp* from *ap-Gal4* could suppress expression of Col, Dimm and Nplp1 (*Benito-Sipos et al., 2011*). Svp is initially expressed by all four Ap cluster cells (St15-16), but is rapidly confined to Tv2 and Tv3, where it is maintained into AFT (*Benito-Sipos et al., 2011*). The reasons for the maintenance of Svp expression in Tv2 and Tv3 were previously not addressed. We find that in *sqz* and *nab* mutants, at stage AFT, there is loss of *svp* expression in one cell and also one extra Dimm expressing cell (*Figure 4F–J*). Conversely, *nab* misexpression results in Svp expression in one extra cell coupled with loss of Dimm expression in one cell (*Figure 4N–Q*). Hence, *sqz* and *nab* control Ap cluster diversification also by ensuring Svp expression in Tv2 and Tv3.

We next addressed the connection between *Kr* and *svp*. We found that in *Kr* mutants, Svp is expressed in one additional Ap cluster cell. The coincidental loss of Dimm and Nplp1 expression leads us to conclude Svp is activated in the Tv1 cell (*Figure 5A–E*). Conversely, *Kr* misexpression postmitotically in all four Ap neurons resulted in complete loss of Svp expression (*Figure 5F–H*).

These findings prompted us to address if the role of *Kr* as a repressor of *svp* is a more general feature during VNC development. To this end, we utilized a *ham-Gal4* driver (*GMR80G10-GAL4*). This driver expresses in several NBs of rows 3 and 4, and hence does not overlap with NBs of row 5 and 6, including NB5-6T, evident by its non-overlap with the row 5 and 6 marker Gsb-n (*Figure 5— figure supplement 1*). The *ham-Gal4* lineages contain Kr and Svp expressing cells, but similarly to NB5-6T, these are non-overlapping (*Figure 5—figure supplement 1*). This provided a scenario to test for the repressive action of *Kr* onto Svp outside the NB5-6T lineage. Misexpression of *Kr* under the control of *ham-Gal4* resulted in striking and significant loss of Svp expressing cells in *ham*-lineages (*Figure 5—figure supplement 1*). To address if *Kr* is also necessary for Svp repression in these lineages, we placed the *ham-Gal4* reporter line into a *Kr* mutant background. While we observed a trend of increased numbers of Svp expressing cells in *Kr* mutants, this was not significant (*Figure 5— figure supplement 1*). However, we noted an apparent increase in Svp expressing cells in the region in between the *ham-Gal4* lineages (*Figure 5—figure supplement 1*), and quantification of this region did reveal a significant increase in Svp expressing cells in *Kr* mutants (*Figure 5—figure supplement 1*). We conclude that the role of *Kr* as a repressor of *svp* extends into many, albeit not all, NB lineages in the developing VNC.

Analyzing *svp* mutants, we observed ectopic Kr expression within the Ap cluster (*Figure 5I–K*). However, when we misexpressed *svp* from *elav-Gal4*, while we could reproduce the complete loss of Dimm expression from the Ap cluster, we found no loss of wildtype Kr expression in Tv1 (*Figure 5L– O*). Prompted by these findings, we analyzed *svp, Kr* double mutants. We found that double mutants showed the typical extra expression of Dimm and Nplp1 observed in *svp* mutants (*Figure 5P–T*).

These results demonstrate that *sqz-nab* ensures maintained expression of Svp in Tv2/3, by repressing Kr. *svp* in turn acts as a potent repressor of *col* and *dimm* expression, and is necessary but not sufficient to repress *Kr*. The key role *Kr* plays in the Tv1 cell is to suppress Svp expression, to safeguard the expression of *col* and *dimm*, and thereby ensure the propagation of the *col>ap/eya>- dimm>Nplp1* feedforward cascade leading to Tv1/Nplp1 terminal differentiation.

## *Kr* can act globally to refine the *ladybird early* and *collier* combinatorial code by repressing Svp

We find that the main role of *Kr* with regards to specifying Tv1 fate is to repress *svp*. Is this interaction restricted to NB5-6T and the Ap cluster, or is it a global phenomenon? To address this we misexpressed *Kr* pan-neuronally, using *elav-Gal4*. Intriguingly, we find that *Kr* strongly suppresses Svp expression broadly in the VNC (*Figure 6A–B, E*).

We recently found that *lbe* and *col* can act in a potent combinatorial code to activate Ap cluster cell fate throughout the VNC (*Gabilondo et al., 2016*). Indeed, co-misexpressing *lbe* and *col* together result in striking ectopic Ap and Eya expression (*Figure 6C,L*). However, only a subset of these ectopic Ap cluster cells express Dimm and Nplp1, and are *bona fide* Tv1 neurons (*Figure 6L*). This observation, together with previous studies (*Baumgardt et al., 2009*; *Baumgardt et al., 2007*; *Gabilondo et al., 2016*), indicate that while *lbe* and *col* are sufficient to activate an early 'generic' Ap cluster cell fate, with ectopic Ap and Eya, specification of the different Ap cluster cell fates i.e.,

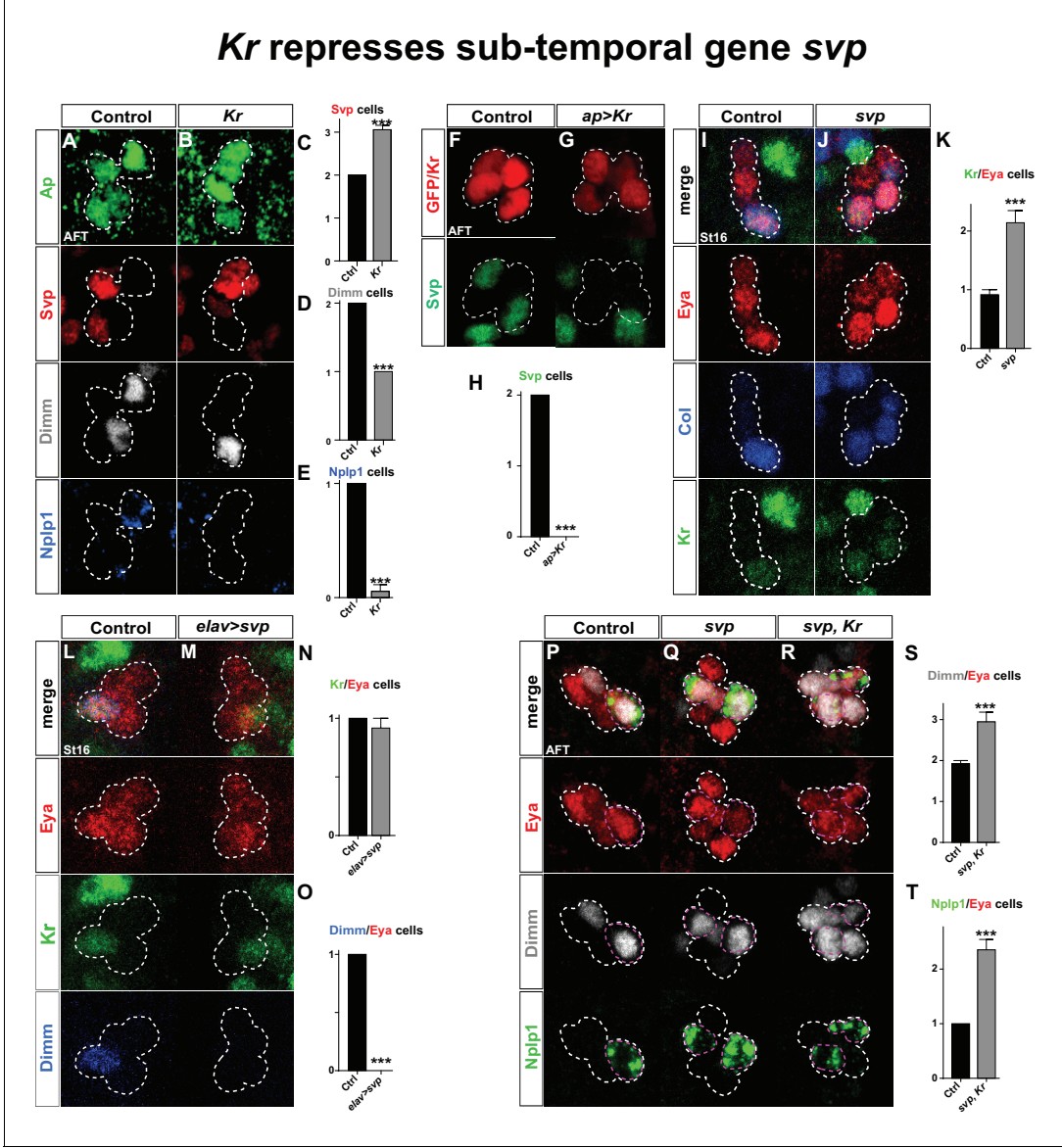

**Figure 5.** *Kr* represses the sub-temporal gene *svp*. (**A–B**) *Kr* mutants at AFT, reveals ectopic Svp expression in the Tv1 cell, accompanied by loss of Dimm expression. (**C–E**) Quantification of Svp/Ap, Dimm/Ap and Nplp1/Ap expressing cells (***p≤0.001; n ≥ 16 clusters; Chi-square test ± SEM). (**F–G**) Misexpression of *Kr* from the late and Ap cluster specific driver *ap-Gal4,* results in loss of Svp expression from Ap2/3, at AFT. (**H**) Quantification of Svp/ GFP or Svp/Kr expressing cells (***p≤0.001, n ≥ 12 clusters, Chi-square test). (**I–J**) *svp* mutants show increased number of Kr expressing cells in the Ap cluster. (**K**) Quantification of Kr/Eya expressing cells (***p≤0.001, n ≥ 12 clusters, Student's *t*-test ± SEM). (**L–M**) Misexpression of *svp* from *elav-Gal4* suppresses Dimm expression, but has no effect on Kr. (**N–O**) Quantification of Kr/Eya and Dimm/Eya expressing cells (***p≤0.001; n ≥ 12 clusters; Chi-square test ± SEM). (**P–R**) Both *svp* and *svp,Kr* double mutants show an increase of Dimm and Nplp1 expressing cells in the Ap cluster. (**S–T**) Quantification of Dimm/Eya and Nplp1/Eya cells (***p≤0.001, n ≥ 12 clusters, Student's *t*-test ± SEM in (**S**) and Chi-square test +/-SEM in (**T**)). Genotypes: (**A, I, L, P**) *OregonR*. (**B**) *Kr[1], Kr[CD]/Kr[1], Kr[CD]*. (**F**) *ap-Gal4/+;UAS-nmEGFP/+*. (**G**) *ap-Gal4/UAS-Kr; UAS-Kr/+*. (**J, Q**) *svp[1]/svp[1]*. (**M**) *elav-Gal4/ UAS-svp*. (**R**) *Kr[1], Kr[CD]/Kr[1], Kr[CD]; svp[1]/svp[1]*.
The following figure supplement is available for figure 5:

**Figure supplement 1.** Kr regulates Svp expression in other NB lineages than NB5-6T.

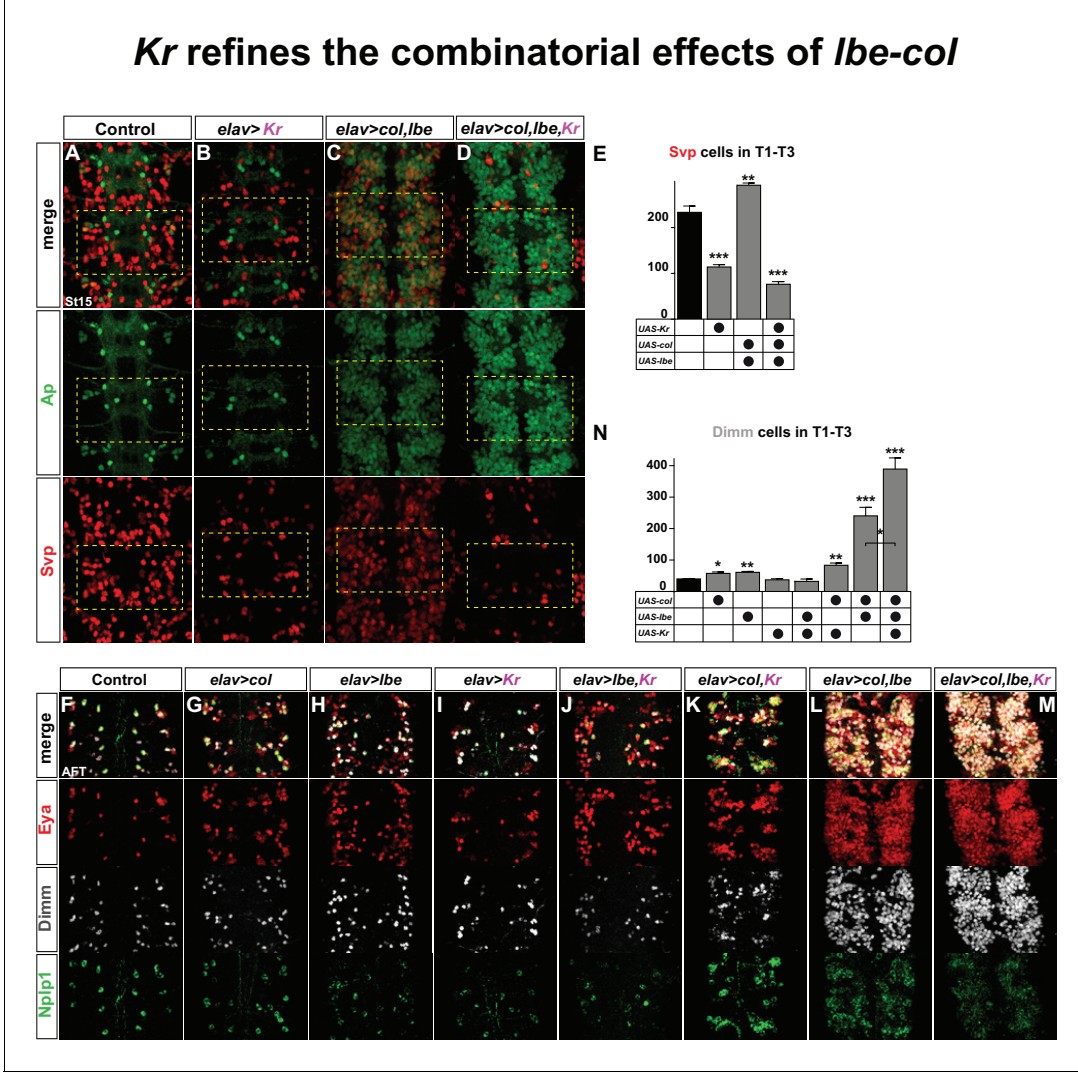

**Figure 6.** *Kr* refines the combinatorial effects of *lbe-col*. (A–B) Single misexpression of *Kr* results in reduced numbers of Svp positive cells in thoracic segments T1-T3 compared to control. (C) Combinatorial misexpression of *col* with *lbe* results in ectopic Ap and Svp expressing cells. (D) Triple misexpression of *col*, *lbe* and *Kr* still results in ectopic Ap cells, but the number of Svp expressing cells is significantly reduced when compared to control. (E) Quantification of Svp expressing cells (**p≤0.01; ***p≤0.001; Student's *t*-test; n = 3 thoracic regions per group; ±SEM). (F–M) Single, double and triple co-misexpression of *col*, *lbe* and *Kr* at AFT from *elav-Gal4*. (F–H) Single misexpression of *col* or *lbe* results in ectopic Eya, Dimm and Nplp1 cells, when compared to control. (I–J) Single misexpression of *Kr* or double misexpression with *lbe* results in ectopic Eya and Nplp1 expression, but not of Dimm, when compared to control. (K) Co-misexpression of *col* and *Kr* results in ectopic Eya, Dimm and Nplp1 cells. (L) Co-misexpression of *col* and *lbe* results in extensive ectopic Eya, Dimm and Nplp1 expression. (M) Triple co-misexpression of *col*, *lbe* and *Kr* results in ectopic Eya, Dimm and Nplp1 cells, with Dimm cell numbers increased in comparison to double misexpression. (N) Quantification of Dimm expressing cells in thoracic segments T1-T3 (*p≤0.05; **p≤0.01; ***p≤0.001; n = 3 thoracic regions; Student's *t*-test +/- SEM). Genotypes: (A, F) *OregonR*. (B, I) *elav-Gal4; UAS-Kr/+; UAS-Kr/+*. (C, L) *elav-Gal4/UAS-col, UAS-lbe*. (D, M) *UAS-Kr; UAS-col, UAS-lbe/elav-Gal4*. (G) *elav-Gal4/UAS-col*. (H) *elav-Gal4/UAS-lbe*, (J) *UAS-lbe/+, UAS-Kr/elav-Gal4*. (K) *UAS-Kr/+, UAS-col/elav-Gal4*.

Tv1/Nplp1, Tv2/3, Tv4/FMRFa requires several other inputs. On this note, *col* and *lbe* co-misexpression also triggered ectopic activation of Svp expression, corroborating the idea that *col* and *lbe* act together to trigger a mixture of Ap cluster cell fates (*Figure 6C,E*). Given the potency of *Kr* in repressing Svp, we speculated that co-misexpression of *lbe* and *col* together with *Kr* may act to suppress the ectopic Svp expression. Indeed, while triple co-misexpression of *Kr*, *lbe* and *col* still activates extensive ectopic Ap expression, as anticipated, the number of Svp expressing cells is dramatically reduced (*Figure 6D,E*). These results suggested that the generation of 'generic' Ap

cluster cell fates triggered by *lbe-col* co-misexpression, could be refined by co-misexpression of *Kr*, and result in a more specific cell fate transformation towards exclusively Tv1/Nplp1 fate.

To test this idea, we misexpressed the three different factors alone or in all combinatorial mixes from *elav-Gal4*, and scored for Eya, Dimm and Nplp1 expression. To quantify the specification of Tv1 cells in contrast to 'generic' Ap cells, we counted the number of Dimm expressing cells. While single misexpression of *lbe* or *col* resulted in increased number of Dimm positive cells compared to control, *Kr* misexpression alone did not result in ectopic Dimm cells (*Figure 6F–I,N*). Co-misexpression of *lbe* and *Kr* did not result in ectopic Dimm cells (*Figure 6J,N*). In contrast, combinatorial misexpression of *col* and *Kr* increased the number of Dimm cells (*Figure 6K,N*). As previously reported (*Gabilondo et al., 2016*), we find that *col* and *lbe* together are tremendously potent at activating ectopic Eya, Dimm and Nplp1 expression in the thoracic region of the VNC, and Dimm cell quantification shows a four-fold increase compared to control (*Figure 6L,N*). Finally, adding Kr to this cocktail refines the *lbe-col* effects, and 'generic' Ap neurons are now exclusively steered towards the Tv1/Nplp1 cell fate, evident by a robust increase in the numbers of Dimm/Nplp1 cells (*Figure 6M, N*).

We conclude that *lbe* and *col* co-misexpression, as anticipated, triggers extensive generation of ectopic Ap cluster cells. However, due to the simultaneous activation of Svp in many ectopic Ap cluster cells, a mixture of Ap cluster cell fates is generated. Adding *Kr* to this misexpression cocktail ensures robust suppression of Svp, and thereby refines the effect of *lbe-col* into exclusively Tv1 cell fate.

## Discussion

Combining the findings presented in this study with previous studies, we find that the temporal gene cascade results in the expression of Cas in the latter part of NB5-6T (*Figure 7A*). *cas* acts together with spatial input, provided by *Antp, hth, exd* and *lbe* to activate *col* in the NB. *col* in turn activates *ap* and *eya* in the early postmitotic cells, which represents a transient and generic Ap cluster cell fate. *col* subsequently acts in a feedforward loop of *col>ap/eya>dimm>Nplp1* to determine Tv1 cell fate. However, in addition to *col*, *cas* activates five other genes, including the last temporal gene *grh*, and the sub-temporal genes *sqz, nab, svp* and, as shown in this study, *Kr*. These five genes engage in a postmitotic cross-regulatory interplay, unique to each of the three cell types, which results in the propagation of the *col>ap/eya>dimm>Nplp1* terminal selector cascade exclusively in Tv1, and the *ap/eya/dac>dimm/BMP>FMRFa* cascade in Tv4, while the Tv2/3 cells acquire a non-peptidergic interneuron identity. The role of *Kr* is to suppress the sub-temporal gene *svp*, in order to safeguard the expression of *col* and *dimm*, and thereby ensures the propagation of the *col>ap/eya>dimm>Nplp1* terminal selector cascade, crucial for specification of the Tv1 cells. The other four genes (*grh, sqz, nab, svp*) each have unique roles, and act as sub-temporal micromanagers to ensure high fidelity and precision in the sub-division of the *cas* temporal window (*Figure 7A*).

### *castor* triggers two FFLs and a sub-temporal program

The temporal gene *cas* plays a pivotal role in the specification process of the different Ap cluster cells due to its activator role on a number of downstream regulators; *col*, a terminal selector in Tv1 specification, the sub-temporal genes *sqz, nab, svp* and *Kr*, as well as the temporal gene *grh*. Strikingly, *cas* thus activates both of the two terminal selector FFLs, and the genes required to refine both FFLs.

*cas* activates *Kr* and *svp*, but how is *Kr* expression then restricted to only Tv1 and *svp* expression to Tv2/3? For *Kr*, restricted expression of *sqz, nab* and *svp* in Tv2-Tv4, all of which suppress *Kr*, can explain the confined expression pattern of *Kr* to Tv1. The gradually restricted expression of *svp* in Tv2-3 is in turn explained by *Kr* repressing *svp* in Tv1, and by *grh* repressing *svp* in Tv4. However, because *grh* misexpression is not sufficient to repress *svp*, it is tempting to speculate that there exists a similar factor to *Kr*, being exclusively expressed in the Tv4 cell, acting to suppress *svp* expression in a highly confined manner to ensure FMRFa/Tv4 specification.

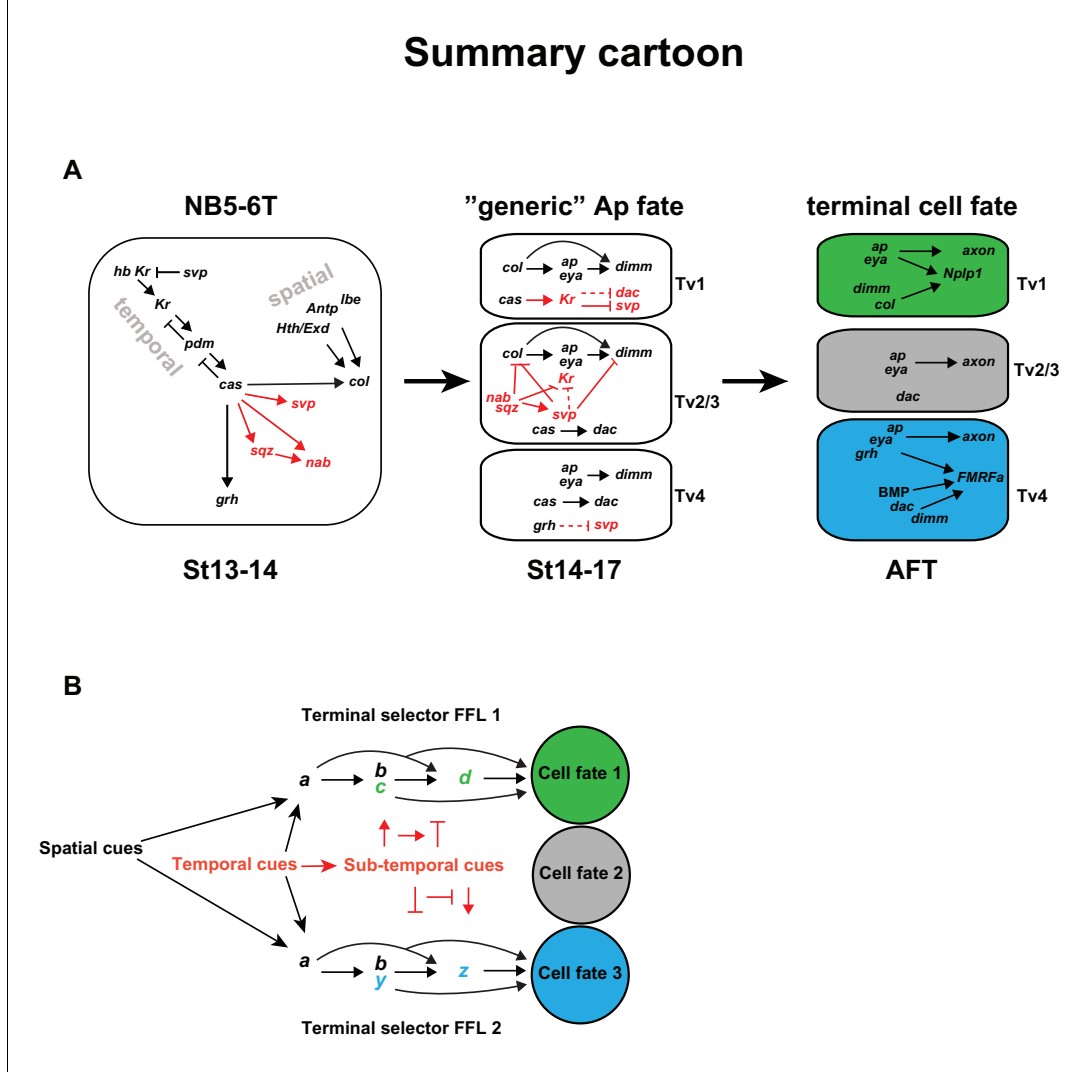

**Figure 7.** Model. (**A**) Cartoon outlining the temporal progression of the NB5-6T and the specification of the Ap cluster cells (based on this and previous studies; see text for references). In the NB, *cas* triggers the sub-temporal genes *sqz, nab, svp* (red) and the temporal factor *grh* (black), and together with spatial input from *Antp, lbe, Hth/Exd* (black), the terminal selector *col*. In the generic Ap cluster cells, the suppressive role of *Kr* on *svp* allows for the *col>ap/eya>dim>Nplp1* FFL to progress. The suppressive roles of the sub-temporal genes *sqz, nab* and svp on *Kr*, allows for continued *svp* expression, which prevents the progression of the Nplp1 FFL. *dac* and *grh* expression in Tv4 ensures the final cell fate of the Tv4/FMRFa neuron. (**B**) Model of how spatial (black) and temporal (red) cues can act to initiate different FLLs (black). In addition, temporal cues can initiate a sub-temporal program (red), which controls the FLL progression (FFL1 and FLL2) in different cell types, by either activating or repressive roles on the terminal selectors (green or blue) crucial for the specification of a distinct cell fate 1 (green) or cell fate 2 (blue).

## Spatial cues trigger FFL, while temporal cues trigger both FFL and sub-temporal genes

Besides its activation by *cas*, *col* activation requires additional spatial information, provided by *lbe*, *Antp*, *hth* and *exd* (*Gabilondo et al., 2016*; *Karlsson et al., 2010*), which subsequently initializes the generic Ap cluster program, by activating *ap* and *eya*. In contrast, *cas* alone activates *grh* and the sub-temporal factors, which are then important for the cell diversification, whether by activating or repressing each other's actions, or the FFLs, or partake in the FFL (*grh*), in order to allocate the correct cell fate to the four Ap cluster neurons. Remarkably, the four spatial inputs (*lbe*, *Antp*, *hth* and *exd*) act only on *col*, while the temporal input (*cas*) acts both on *col*, as well as the temporal and sub-temporal factors (*sqz, nab, svp, Kr* and *grh*). It is tempting to speculate that this may point to a general role for spatial versus temporal cues, and may be logically explained by the fact that spatial

cues generally do not display the highly selective temporal expression profile necessary for sub-temporal cell diversification (*Figure 7B*).

## Expanding roles of temporal genes

An unexpected finding in this study pertains to the dual role of *Kr*, first acting early in the canonical temporal cascade and subsequently late in the sub-temporal cascade, to ensure the specification of the Tv1 cell. The main role of *Kr* in Tv1 cells is to suppress *svp*, hence allowing for the maintenance of Col, which itself is critical for the propagation of the terminal FFL, fundamental for Tv1 cell fate. Interestingly, dual expression of *Kr*, first in the neuroblast and subsequently in neurons, was previously observed in NB3-3, but the functional role of the second *Kr* expression pulse was not addressed (*Tsuji et al., 2008*). *svp* itself also displays a dual expression and function, being expressed early in many NB lineages to suppress *hb* (*Kanai et al., 2005*), then being re-expressed in several linages, and in NB5-6T it acts to suppress *col* and *dimm* (*Benito-Sipos et al., 2011*). With regards to postmitotic activity, another example of a temporal gene acting postmitotically applies to the role of the last temporal gene, *grh*, which is necessary and sufficient for FMRFa expression in Tv4 cells, and can trigger ectopic FMRFa in Ap neurons when misexpressed postmitotically (*Baumgardt et al., 2009*). Yet in contrast to *Kr*, *grh* does not experience a dual expression profile. Hence, with several examples of dual (*Kr* and *svp*) and progenitor versus postmitotic roles of temporal genes (*Kr* and *grh*), it is tempting to speculate that this type of temporal multi-tasking may indeed be a common feature for many temporal genes, both in *Drosophila* and in higher organisms.

# Materials and methods

## Fly stocks

*lbe(K)-Gal4* (*Baumgardt et al., 2009*). *lbe(K)-EGFP* (*Ulvklo et al., 2012*). *elav-Gal4* (*DiAntonio et al., 2001*) (provided by A. DiAntonio). *cas^{Δ1}* and *cas^{Δ3}* (*Mellerick et al., 1992*) (provided by W. Odenwald). *UAS-nls-myc-EGFP* (referred to as *UAS-nmEGFP*) (*Allan et al., 2003*). *UAS-col* (*Vervoort et al., 1999*) (provided by A. Vincent). *grh^{370}* (*Bray and Kafatos, 1991*) (provided by S. Bray). *Kr^1*, *Kr^{CD}* (*Isshiki et al., 2001*) (provided by C.Q. Doe). *nab^{R52}* and *UAS-nab* (*Terriente Félix et al., 2007*) (provided by F. Diaz Benjumea). *Df(lbl-lbe)B44* and *UAS-lbe* (provided by K. Jagla). *Antp^{12}* (*Abbott and Kaufman, 1986*) (provided by F. Hirth).

From Bloomington Drosophila Stock Center: *Antp^{25}* (BL#3020). *lbe^{12C005}* (BL#59385). *elav^{C155}*=*elav-Gal4* (BL#458). *Df(2L)BSC354 (eya^{Df})* (BL#24378). *elav-Gal4* (BL#8765). *sqz^{ie}*(BL#36497). *svp^1* (BL#6190). *ap^{md544}* (referred to as *ap^{Gal4}*) (BL#3041). *grh^{Df3654}*=*Df(2R)Pcl7B* (BL#3064). *sqz^{ie}* (BL#36497). *ham-Gal4 = GMR80G10-GAL4* (BL#40090).

*UAS-col-HA* and *UAS-myc-lbe* were generated by the codon optimization for expression in *Drosophila* (http://www.kazusa.or.jp/codon/). EcoRI site and consensus start codon (*Cavener and Ray, 1991*) was added to the 5' end. Three different stop codons (amb, och, opa) followed by an XbaI site were added to the 3' end (see *Supplementary file 1* for both sequences). DNAs were generated by gene-synthesis (Genscript, New Jersey), and cloned into pUASattB (*Bischof et al., 2007*), as EcoRI/XbaI fragments. DNAs were injected into landing site strains BL#9736 (53B) and BL#9744 (89E), for *UAS-myc-lbe*, and BL#9750 (65B) for *UAS-col-HA* (BestGene, CA).

Mutants were maintained over *GFP*- or *YFP*-marked balancer chromosomes. As wild-type control *OregonR* was used. Staging of embryos was performed according to Campos-Ortega and Hartenstein (*Campos-Ortega and Hartenstein, 1985*).

## Immunohistochemistry

Guinea pig a-Cas antibodies were generated by inserting a DNA fragment coding for Cas amino acids 171–793 in the Cas-PA protein into plasmid pET16b, followed by an expression in bacteria, gel purification and injection into 4 guinea pigs (Davids Biotechnologie, Regensburg, Germany). The strongest sera was used at 1:500, and tested for loss of specific nuclear staining in the VNC in *cas* mutants. Other primary antibodies were: Guinea pig a-Deadpan (1:1000) and rat a-Dpn (1:200) (*Ulvklo et al., 2012*). Guinea pig a-Col (1:1000), guinea pig a-Dimm (1:1000), chicken a-proNplp1 (1:1000) and rabbit a-proFMRFa (1:1000) (*Baumgardt et al., 2007*). Rat a-Grh (1:1000) and rat a-Nab (1:500) (*Baumgardt et al., 2009*). Rabbit a-Cas (1:250) (provided by W. Odenwald). Mouse

mAb a-Dac dac2–3 (1:25), mAb a-Eya 10H6 (1:250) (Developmental Studies Hybridoma Bank, Iowa City, IA, US). Rabbit a-Kr (1:500) (provided by Ralf Pflanz), mAb a-Svp (1:50) (provided by Y. Hiromi). Rabbit a-Ap (1:1000) (*Bieli et al., 2015*) (provided by D. Bieli and M. Affolter). Chicken a-GFP 1:1000 (Abcam, ab13970). Rat monoclonal α-Gsb-n (1:10) (provided by R. Holmgren).

## Confocal imaging and data acquisition

Zeiss LSM 700 Confocal microscopes were used for fluorescent images; confocal stacks were merged using LSM software or Adobe Photoshop. Statistic calculations were performed in Graphpad prism software (v4.03). Cell counts in *Figure 6* were done manually on thoracic segments T1-T3 in 3 VNCs for each group, with ImageJ FIJI and numbers transferred to Graphpad prism. To address statistical significance Student's *t*-test or in the case of invariant cell numbers contingency tables together with Chi-Square test were used. Images and graphs were compiled in Adobe Illustrator.

## Acknowledgements

We are grateful to A DiAntonio, W Odenwald, R Holmgren, A Vincent, S Bray, CQ Doe, F Diaz Ben-jumea, K Jagla, F Hirth, R Pflanz, Y Hiromi, D Bieli, M Affolter, Developmental Studies Hybridoma Bank at the University of Iowa, the Bloomington Stock Center, and the FlyORF stock center for sharing antibodies, fly lines and DNAs. We are grateful to Adrian Moore for bringing it to our attention that the expression of *ham-Gal4* was apparent in subsets of NBs in the developing CNS. We thank Doug W Allan for critically reading the manuscript. H Ekman, C Jonsson and A Starkenberg provided excellent technical assistance.

## Additional information

### Funding

| Funder | Grant reference number | Author |
| --- | --- | --- |
| Svenska Forskningsrådet Formas | 621-2013-5258 | Stefan Thor |
| Cancerfonden | 120531 | Stefan Thor |
| Knut och Alice Wallenbergs Stiftelse | KAW2011.0165 | Stefan Thor |
| Ministerio de Industria, Energía y Turismo | BFU2013-43858-P | Jonathan Benito-Sipos |

The funders had no role in study design, data collection and interpretation, or the decision to submit the work for publication.

### Author contributions

JS, HG, Conception and design, Acquisition of data, Analysis and interpretation of data, Drafting or revising the article; JB-S, ST, Conception and design, Analysis and interpretation of data, Drafting or revising the article

### Author ORCIDs

Stefan Thor, http://orcid.org/0000-0001-5095-541X

## Additional files

### Supplementary files

• Supplementary file 1. DNA sequences. DNA sequences for new UAS constructs.

• Supplementary file 2. Quantification data. Raw cell counts for the graphs in this study.

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
