## [Decision Letter]

Thank you for submitting your article "Temporal Diversification of Neuronal Cell Fate by Sub-temporal Micromanagement" for consideration by *eLife*. Your article has been reviewed by three peer reviewers, and the evaluation has been overseen by a Reviewing Editor (Robb Krumlauf) and Marianne Bronner as the Senior Editor. The following individuals involved in review of your submission have agreed to reveal their identity: Tzumin Lee (Reviewer #3).

The reviewers have discussed the reviews with one another and the Reviewing Editor has drafted this decision to help you prepare a revised submission.

It is the consensus opinion of the reviewers that this is a well written paper with clear experiments that continues detailed and elegant work by the group on the mechanisms generating neuronal diversity within the NB5-6 lineage of the *Drosophila* embryonic ventral nervous system. The identification of temporal transcription factors has allowed an understanding of how neuronal diversity is established during development. However, many lineages display broad temporal windows in which different cell types are produced. It is unclear how such specification can be achieved with so few temporal transcription factors (TTFs). In the current paper, the authors address this question and present three novel findings:

1) the late temporal transcription factor (TTF) Castor activates two different cell determination feed forward loops (FFLs), plus one subtemporal regulatory module, to specify the Tv1 and Tv4 neurons

2) the *Kr* early TTF is re-expressed in the postmitotic Tv1 neuron, where it is necessary and sufficient (roughly) to repress Svp, allowing expression of Col, and thus the specification of Tv1 identity.

3) the spatial factors (*lbe, hth, Antp*) act on the cell determination FFLs and not on the subtemporal module.

These results are in line with the authors' previous findings and contribute to a deep understanding of how sub-temporal windows are regulated. However, there are two major comments that must be addressed before this work can be considered for publication:

1) There is a need to better characterize the effects of *Antp/lbe* loss-of-function on *Kr* expression by performing NB5-6-specific RNAi for *Antp* to clarify the phenotype (which is messy in whole animal mutants). It would be valuable to see *lbe-gal4* and *ap-gal4* driving RNAi constructs for *Antp* and *lbe* to clarify their role in regulating *Kr* in the NB5-6 lineage. It would be possible to assay in L1 or even L2 if it takes time for the knockdown to occur.

2) Test the role of *Kr* and Svp late expression through analysis in a second NB lineage to generalize the beautiful results found in the NB5-6 lineage.

The reviewers feel these experiments could both be done quickly and significantly strengthen the paper.

---

## [Author Response]

1) There is a need to better characterize the effects of Antp/lbe loss-of-function on Kr expression by performing NB5-6-specific RNAi for Antp to clarify the phenotype (which is messy in whole animal mutants). It would be valuable to see lbe-gal4 and ap-gal4 driving RNAi constructs for Antp and lbe to clarify their role in regulating Kr in the NB5-6 lineage. It would be possible to assay in L1 or even L2 if it takes time for the knockdown to occur.

We fully agree with the reviewers that the connection between *Antp, lbe* and *Kr* in Ap cluster neurons was not fully resolved. However, the main problem here is actually not that *Antp* and *lbe* mutants are malformed, or that these two genes appear to broadly affect CNS development. In fact, based upon our previous findings, *Antp* and *lbe* mutants develop into late embryonic stages without major malformations. Previously published figures show the late embryonic VNCs of both mutants, stained for Nplp1 (Karlsson et al., PLOS Bio., 2010; Gabilondo et al., PLOS Bio, 2016). The VNCs are fully developed, and while they show a specific loss of the thoracic Ap cluster markers, the dorsal Ap cells are fully specified and express Nplp1

More to the point, it was the loss of Ap cluster markers that prevented us from assessing the identity of the Tv1 cell in the *Antp* and *lbe* mutants. Even with the successful use of RNAi, in later stages of development (L1, L2), we would still encounter this loss of Ap cluster markers, preventing us from identification of the Tv1 cell by the use of the previously used markers for Tv1 identification i.e., Eya, Col, Dimm, Nplp1. To circumvent this problem, we decided to use a different (and by *Antp* or *lbe* mutants unaffected) combination of Ap cluster markers in order to identify the Tv1 cell: Quadruple stains for lbe-GFP, Cas, Nab and *Kr* (made possible by a novel guinea pig anti-Cas antibody generated by us; now described in the Methods section). Using this combination, we find that *Kr* expression is unaffected in the Tv1 cells in both mutants. We decided to replace the previous lower resolution whole lineage analysis in Figure 3—figure supplement 1 with this more precise analysis of the Ap cluster, and have changed the main text and the figure legends accordingly (subsection “Postmitotic Kr expression is activated by the late temporal gene *castor”*, second paragraph).

2) Test the role of Kr and Svp late expression through analysis in a second NB lineage to generalize the beautiful results found in the NB5-6 lineage.

The reviewers feel these experiments could both be done quickly and significantly strengthen the paper.

This is also an important point. To address this we utilized a Janelia *Gal4* driver line that we have recently identified as being expressed in one row 3 and two row 4 NBs; the *ham- Gal4* driver. We find non-overlapping Kr and Svp expression in one of the row 4 lineages.

Misexpression of *Kr* from *ham-Gal4* leads to significant reduction of Svp positive cells in this lineage. In addition, we observed significant increase in Svp positive cells in the regions intermediate to the *ham-Gal4* lineages in *Kr* mutants. Hence, the role of *Kr* as a repressor of Svp extends beyond the NB5-6T lineage, albeit not to all VNC lineages. We have compiled these new experiments into a new supplemental figure (Figure 5—figure supplement 1), and changed the main text (subsection “The main role of *Kr* is to suppress the sub-temporal gene *svp*”, third paragraph) accordingly.